# LLaPA: Harnessing Language Models for Protein Enzyme Function

## Abstract

Identifying protein enzyme functions, crucial for numerous applications, is challenging due to the rapid growth in protein sequences. Current methods either struggle with false positives or fail to generalize to lesser-known proteins and those with uncharacterized functions. To tackle these challenges, we propose LLaPA: a Protein-centric Large Language and Protein Assistant for Enzyme Commission (EC) number prediction. LLaPA uses a large multi-modal model to accurately predict EC numbers by reformulating the EC number format within the LLM self-regression framework. We introduce a dual-level protein-centric retrieval: the *protein-level* retrieves protein sequences with similar regions, and the *chemical-level* retrieves corresponding molecules with relevant reaction information. By inputting the original protein along with the retrieved protein and molecule into the LLM, LLaPA achieves improved prediction accuracy, with enhanced generalizability to lesser-known proteins. Evaluation on three public benchmarks show accuracy improvements of **17.03%**, **9.32%**, and **38.64%**. These results highlight LLaPA's ability to generalize to novel protein sequences and functionalities. Codes are provided in the supplement.

## 1 Introduction

Understanding the functions of protein enzymes is crucial for unraveling metabolic pathways (Fonseca et al., 2019), diagnosing diseases (Hewitt et al., 2004; Voller et al., 1976), advancing personalized medicine (Sookoian & Pirola, 2015), facilitating industrial applications (Victorino da Silva Amatto et al., 2022; Bernal et al., 2018; Chapman et al., 2018; Basso & Serban, 2019), understanding biological evolution (Campbell et al., 2016), and beyond. Recently, advances in biological technologies have unveiled a vast array of enzyme protein sequences from organisms spanning the entire tree of life. However, only a small fraction of the protein has been manually annotated (i.e., $\sim 0.3\%$ (Boutet et al., 2007) in UniProtKB (The UniProt Consortium, 2023) is manually annotated.) The computational methods can bridge the sequence-annotation gap, but the critical assessment of protein function annotation (CAFA) study found that $\sim 40\%$ of the computation annotations are incorrect (Radivojac et al., 2013). Additionally, there exists a portion of proteins that are not similar enough to any characterized protein to infer function and their function remains unknown (Price et al., 2018a). Therefore, the functional annotation of understudied and promiscuous proteins remains an overwhelming challenge in protein science (Jeffery, 2018; Hult & Berglund, 2007).

In the past few years, the enzyme function annotation has been formulated as a multi-label classification tasks (Gligorijević et al., 2021; Lin et al., 2022; Ryu et al., 2019; Sanderson et al., 2023; Dalkiran et al., 2018), aiming to predict the Enzyme Commission (EC) number of annotated enzymes (Webb & International Union of Biochemistry and Molecular Biology, 1992). The EC number is a classification ontology for the chemical reactions catalyzed by enzymes. However, the multi-label classification paradigm suffers from the limited and imbalanced training dataset. Recently proposed CLEAN framework shows the retrieval-based framework can significantly surpass classification deep learning frameworks, such as ProteInfer (Sanderson et al., 2023), DeepEC (Ryu et al., 2019), and DEEPre (Li et al., 2018). Notably, it exhibits remarkable performance on EC numbers represented by fewer than ten sequences, highlighting the superiority of contrastive learning over multi-label classification in predicting enzyme function. However, the framework is not engineered to generalize to proteins with novel functionalities, requiring a certain number of proteins with annotated EC numbers to maintain its generalizability. There are pioneers (Xu et al., 2023b; Gane et al.) aiming to

harness the generalizability of LLM and combine LLM with a protein encoder to create an end-to-end trained large multi-model model for various protein-related tasks. Despite these advancements, their approach primarily emphasizes linking proteins with textual data, often overlooking biological priors. This oversight restricts the model's ability to offer interpretations from a biological standpoint—an aspect that is essential for advancing biological research.

In this paper, we introduce `LLaPA`, a protein-centric, framework for multi-modal large language models (MLLMs) training and inference. In detail, `LLaPA` enhances MLLMs for protein enzyme understanding from two perspectives. ❶ **Focusing on the Natural Language Prior**, we first observed that the LLM struggles to directly and accurately output EC numbers (*i.e.*, "EC 3.4.11.4") due to their specific format—four numbers separated by periods. To counteract this limitation, we redesigned the EC number format by replacing the period with another symbol that is distant from numbers in the embedding space. ❷ **Embracing the Biological Prior**, we build a two-tiered protein-centric retrieval engine, grounded in two fundamental biological insights: (1) *At the protein-level*, recognizing the evolutionary conservation of functionally critical regions within protein sequences, our engine retrieves a protein with similar regions as the reference to infer the query enzyme's function. (2) *At the chemical level*, acknowledging the intrinsic link between an enzyme's catalytic actions and its function, we leverage the retrieved protein to further identify a corresponding molecule. This molecule acts as an additional reference point, refining our EC number prediction capabilities. By querying a protein along with two retrieved entities - a protein and a molecule, `LLaPA` directly predicts the corresponding Enzyme Commission numbers. Our contributions are summarized below:

* ⋆ We introduce `LLaPA` framework, a cutting-edge framework specifically designed for protein enzyme function prediction. `LLaPA` stands out by addressing the unique challenges in protein enzyme function annotation through innovative training and inference strategies tailored for multi-modal large language models (MLLMs).

* ⋆ We identified how the traditional format of EC numbers can be problematic for accurate predictions by large language models (LLMs). To address this, `LLaPA` introduces a new encoding scheme that replaces periods with symbols that are more distinct in the embedding space. This subtle change significantly improves EC number prediction accuracy, indicating the format's better compatibility with the LLMs' self-regression paradigm.

* ⋆ `LLaPA` advances the field with its two-tiered retrieval engine, deeply rooted in biological insights. This engine not only identifies proteins with evolutionary conserved, functionally critical regions but also pairs these proteins with corresponding molecules. This dual approach enhances the prediction of Enzyme Commission numbers, leveraging biological priors at both the protein and chemical levels to refine the model's predictive accuracy.

* ⋆ Our extensive testing across four public datasets confirms the effectiveness of our approach. For example, `LLaPA` achieves $\{17.03\%, 9.32\%, 38.64\%\}$ performance improvements on `Halogenase`, `Price`, and `New` datasets over previous state-of-the-art (`SOTA`) approaches.

## 2 RELATED WORK

**Large Language Model** Large Language Models (LLMs) have demonstrated considerable potential in biology by leveraging vast biological datasets to advance research and understanding. Genomic models such as BioBERT (Lee et al., 2020) and DNABERT (Ji et al., 2021) excel in sequence annotation and gene function prediction. In proteomics, models like ESM-1b (Rives et al., 2019) improve protein sequence understanding, and TAPE (Rao et al., 2019) facilitates evaluation efficiency by providing a standardized benchmark. In drug development, AlphaFold3 (Callaway, 2024) proved superior in finding new drugs. Others, such as SciBERT (Beltagy et al., 2019), a leading language model, significantly improve the extraction and summarization of essential information. Recent research focuses on integrating multimodal data (Zhang et al., 2023a) (Wang et al., 2024). For example, LLaVA (Liu et al., 2024), which connects a vision encoder and an LLM, is the first attempt to extend instruction-tuning to the language-image multimodal space. The BLIP (Li et al., 2022) has demonstrated impressive performance in vision-language tasks and also achieved state-of-the-art zero-shot performance when the models are directly applied to two video-language tasks. In addition, enhancing model interpretability (Joshi et al., 2021) (Nhlapho et al., 2024), and improving prediction robustness (Yang et al., 2023) also attract a lot of attention.

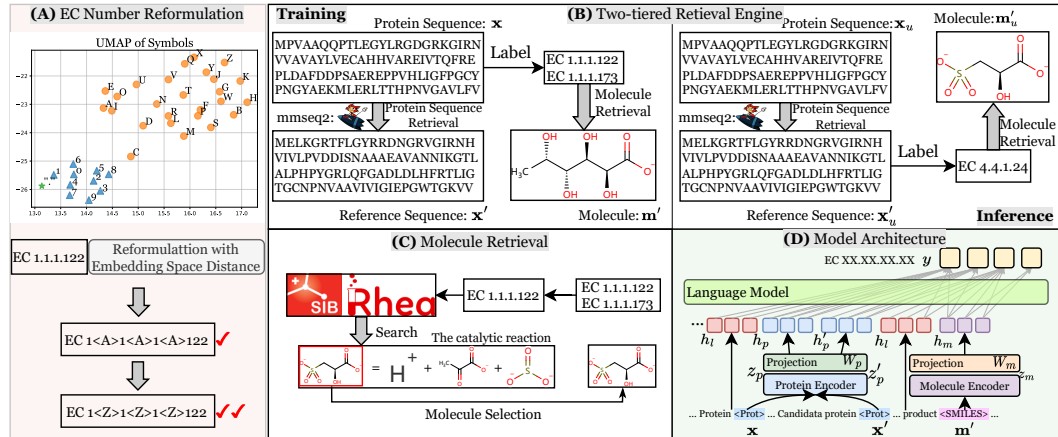

Figure 1: The overview of LLaPA. (**A**) EC Number reformulation. We reformulate the EC number by analyzing the distribution of symbols within the embedding space, adopt the use of LLM self-regression for EC number prediction. (**B**) During training and inference, it employs two-tiered retrieval engine that encompasses both protein sequence and molecule retrieval for accurate EC number prediction. (**C**) For molecule retrieval, we utilize an expert-curated knowledge base. (**D**) All gathered information, along with the query protein, is then processed by an LLM to generate the final prediction.

**Enzyme Function Prediction**  Enzyme function prediction plays a crucial role in the field of biology. Several ways have been devised to forecast enzyme function, such as those relying on sequence similarity (Zhang et al., 2017) (Desai et al., 2011) (Altschul et al., 1997), structural similarity (Altschul et al., 1990), and protein homology (Zhang et al., 2017). InterPro (Paysan-Lafosse et al., 2022) signatures, position-specific scoring matrices (cheol Jeong et al., 2010), pseudo-amino acid composition (Chou, 2009), and machine learning techniques (Amidi et al., 2017) such as multi-label k-nearest neighbour (Huang et al., 2007) and SVM (Mohammad & Nagarajaram, 2011) are all good ways to figure out what multi-functional enzymes do. Furthermore, the deep learning frameworks that integrate representation learning and classifier learning have shown significant promise in enzyme function prediction, such as Proteinfer (Sanderson et al., 2023), DeepEC (Ryu et al., 2019), and DEEPre (Li et al., 2018). A new paradigm was recently introduced by ProTranslator (Xu & Wang, 2022). It deems the process of using function descriptions to predict the amino acid sequence a machine translation problem. This pattern was later expanded with a framework for multilingual translation (Xu et al., 2023a). Additionally, (Yu et al., 2023) introduces a metric learning framework designed to increase the distance between protein embeddings of differing functions and decrease it for those with similar functions, achieving state-of-the-art (SoTA) performance. However, their approach relies solely on a simple triplet loss for contrasting samples and does not integrate biological priors to enhance generalization for functions without a defined EC number.

## 3 METHODOLOGY

**Overview**  LLaPA is a framework designed specifically for predicting the function of protein enzymes, outputting the Enzyme Commission number based on the given protein sequence. First of all, we reformulate the $y$ that is more friendly for LLM prediction (Section 3.1). Then, for a protein sequence $\mathbf{x}$ with $n$ amino acids, LLaPA initially uses $\mathbf{x}$ to identify a reference protein sequence $\mathbf{x}'$, then retrieves the corresponding molecule $\mathbf{m}'$ related to the catalytic reaction involve $\mathbf{x}$ (Section 3.2). As a result, LLaPA employs $\mathbf{x}$, $\mathbf{x}'$, and $\mathbf{m}'$ to predict the functional annotation $y$ of $\mathbf{x}$ (Section 3.3).

Specifically, LLaPA inference adopts a similar design to LLaVA (Liu et al., 2023b;a). With $\mathbf{x}$, retrieved protein $\mathbf{x}'$ and retrieved molecule $\mathbf{m}'$, LLaPA first apply the pre-trained protein encoder $E(\cdot)$ to provide protein features $\mathbf{z}_p = E(\mathbf{x})$ and $\mathbf{z}'_p = E(\mathbf{x}')$. Next it uses the pre-trained molecular encoder $C(\cdot)$ to obtain molecular features $\mathbf{z}_m = C(\mathbf{m}')$. To process these features further, LLaPA uses two projectors: $\mathbf{W}_p$, which converts $z_p$ and $z'_p$ into language embedding tokens $h_p$ and $h'_p$, and $\mathbf{W}_m$, which transforms $\mathbf{z}_m$ into language embedding tokens $h_m$. These projectors map information

from proteins and molecules into the language token space, bridging biological and chemical prior for protein enzyme function understanding. Finally, the query protein $x$, retrieved protein $x'$ and molecule $m'$ along with corresponding instructions can combine together to obtain the EC number of the query. For more details about overall pipeline, please refer to Algorithm 1.

## 3.1 ENZYME COMMISSION NUMBER REFORMULATION

In this section, we introduce the Enzyme Commission (EC) number reformulation. Protein functional annotations, including EC numbers and Gene Ontology (GO) terms, exhibit hierarchical structures. Especially, the Enzyme Commission (EC) number serves as a numerical system for classifying enzymes according to the chemical reactions they facilitate. Within this enzyme nomenclature system, each EC number has four digital numbers, that correspond to a recommended name for the specific enzyme-catalyzed reaction it denotes. As shown in Figure 2, from left to right, the digits correspond to the reaction class, subclass, and sub-subclass, and a serial number that is substrate-specific.

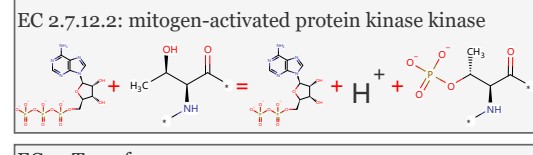

EC **2**: Transferases

EC 2.**7**: Transferring phosphorus-containing groups.

EC 2.7.**12**: Dual-specificity kinases (those acting on Ser/Thr and Tyr residues).

EC 2.7.12.**2**: mitogen-activated protein kinase kinase, activate MAP kinases through dual phosphorylation on serine/threonine and tyrosine residues.

Figure 2: A case of EC number format.

However, we've observed that while Large Language Models (LLMs) struggle to predict the EC number for given protein sequences. *We suspect this limitation may be rooted in the characteristics of the embedding space.* To explore this hypothesis, we began by visualizing the embedding of symbols, including numbers, letters, and the "." character. As shown in Figure 1 (**A**), we noticed that the "." character is positioned closely to the numbers in the embedding space. This proximity suggests that predicting the EC number may be akin to predicting a single, large numerical value. Accurately predicting such a large number presents a significant challenge (Yuan et al., 2023; Zhang et al., 2020; Sundararaman et al., 2020; Jin et al., 2024).

Therefore, we first replaced the "." with the letter "A", and we got an improvement for predicting the EC number. Then we further replace "A" with "Z" which is farther away from numbers in the embedding space, and then get further improvement. Please refer to Section 4 for a detailed discussion of the demonstration results.

After reformulating the Enzyme Commission (EC) number for large language model (LLM) predictions, we are now able to accurately predict the first three digits of the EC number. This outcome suggests that the model is capable of understanding protein functions but falls short in identifying the specific catalytic reaction utilized by the protein, *i.e.*, correctly predicting the four digits of the EC number. To address this limitation, we require further reference information to assist the model in pinpointing the precise catalytic reaction associated with the protein.

## 3.2 INTEGRATING BIOLOGICAL PRIOR KNOWLEDGE BY RETRIEVAL ENGINE

In this section, we introduce a novel two-tiered retrieval engine, a cornerstone of `LLaPA` integrates biological prior knowledge to prompt LLMs to predict the four digits of the EC number. This engine is divided into two specialized modules: the first addresses the retrieval of reference protein sequences, while the second concentrates on the identification of molecules relevant to chemical reactions.

**Protein Prior Knowledge Module - Retrieval of Reference Protein Sequences.** A fundamental principle in understanding protein function is that regions of protein sequences important for function tend to be conserved through evolution. Consequently, proteins sharing similar regions are likely to possess similar enzymatic functions and may even catalyze the same reactions. Inspired by this insight into protein functionality, we employ "mmseq2" (Steinegger & Söding, 2017), a comprehensive software suite designed for the efficient searching and clustering of extensive protein and nucleotide sequence datasets based on significant protein-related knowledge. This tool enables us to identify

the most closely related protein sequence as a reference, thereby aiding the model in accurately predicting the four digits of the EC number. When given a protein, `LLaPA` utilizes "mmseq2" to find the most similar protein sequence, $\mathbf{x}'$, within a specified protein database.

Specifically, input the query protein sequence $\mathbf{x}$ to the "mmseq2". It will search in the specified protein database and output the $\mathbf{x}'$ with the highest sequence identify cutoff value in the protein database.

**Chemical Reaction Prior Knowledge Module - Retrieval of Corresponding Molecules.** The simplest way to identify a catalytic reaction is by examining the reaction itself. This module is designed to retrieve a molecule in SMILES format [1] that participates in the catalytic reaction associated with a given protein. Yet, the task of retrieving the catalytic reaction based solely on the protein sequence is exceedingly difficult. Fortunately, the "Protein Prior Knowledge Module" presents an opportunity to bypass the direct retrieval of molecules by protein sequence. Therefore, we employ the "rhea" (Bansal et al., 2022), an expert-curated knowledgebase of chemical and transport reactions of biological interest, and the standard for enzyme and transporter annotation in UniProtKB. Noticeably, the "rhea" necessitates the EC number to fetch the relevant catalytic reaction—the very information we aim to predict.

During the training phase, as shown on the left side of Figure 1 (**B**), we directly input the EC number (*i.e.*, the label) of the protein sequence to the "Molecule Retrieval" module. During the inference phase (the right side of Figure 1 (**B**)), the EC number of the input protein sequence $\mathbf{x}_u$ is unavailable. Therefore, we first input the protein sequence $\mathbf{x}_u$ to the "Protein Prior Knowledge Module" and get a protein sequence $\mathbf{m}'_u$. The protein sequence $\mathbf{m}'_u$ from the protein database has EC numbers. Consequently, we fed its EC numbers into the "Molecule Retrieval" module.

As depicted in Figure 1 (**C**), our "Molecule Retrieval" module operates as follows: (1) it randomly selects one EC number from the input EC numbers; (2) it inputs the selected EC number into the "rhea", which then outputs the corresponding catalytic reaction; (3) it selects the first reactant molecule in the catalytic reaction to be the output molecule $\mathbf{m}'$.

We emphasize that the retrieve logic in the inference phase is reasonable, as proteins with high sequence identify cutoff values typically exhibit similar enzyme functions Gerlt et al. (2015); Yu et al. (2023). Therefore, their molecules in the corresponding chemical enzyme reactions should possess similar catalytic information. For instance, the protein "T1RRJ4" and its corresponding retrieved protein "Q2XSC6" have the EC numbers "EC 4.2.3.10" and "EC 4.2.3.20", respectively. Interestingly, the first reaction molecule for both is "(2E)-geranyl diphosphate". As depicted in Figure 1 (**C**), our "Molecule Search" process randomly chooses an EC number when multiple are available; if only one EC number exists, that EC number is utilized.

### 3.3 MODEL ARCHITECTURE AND TRAINING

In this section, we delve into the details of the network architecture designed to underpin the proposed retrieval engine, the corresponding multi-modal training pipeline, and the techinical details of `LLaPA`. We also illustrate the flow of data and the trainable parameters during training in Figure 7(a).

**Network Architecture** The network architecture is illustrated in Figure 1 (**D**). `LLaPA` features several key components: an LLM for natural language processing, a protein encoder, and a corresponding projector that bridges the protein encoder with the LLM. Additionally, it includes a molecule encoder and its own projector to link the molecule encoder with the LLM. We use Vicuna-7b (Zheng et al., 2023) as the backbone of `LLaPA`, which is a chat assistant trained by fine-tuning Llama 2 on high-quality dialog

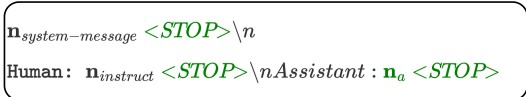

Figure 3: The input sequence employed to train the model is designed to teach the model to predict the assistant's responses and to determine the appropriate point to conclude. Consequently, only the green sequence/-tokens are utilized in calculating the loss within the auto-regressive model.

---

[1] The simplified molecular-input line-entry system (SMILES) is a specification in the form of a line notation for describing the structure of chemical species using short ASCII strings. (Weininger, 1988)

datasets. To make `LLaPA` understand protein sequences (*i.e.*, sequence of amino acid tokens, which are the primary structure of proteins), we employ ESM-2 (Lin et al., 2022) as the protein encoder $E(\cdot)$, the general-purpose protein language model. For the molecule, we use ChemBERTa (Chithrananda et al., 2020) as the molecule encoder $C(\cdot)$, a language model pre-trained on a chemical dataset called PubChem (Kim et al., 2019) that consists of molecules in SMILES format. These two projectors that connect the protein feature, and molecule feature into language embedding tokens $h_l$ are both a two-layer MLP.

**Multi-modal Training** For each protein $\mathbf{x}$, we create a single-turn conversation dataset $(\mathbf{n}_q, \mathbf{n}_a)$. These are arranged sequentially, with the answers interpreted as the assistant's responses and the question as the instruction, denoted as $\mathbf{n}_q$. This arrangement follows a unified format for multimodal instruction-following sequences, as illustrated in Figure 3. We set the trainable parameters as $\theta$, $\mathbf{n}_{instruct,<i}$ and $\mathbf{n}_{a,<i}$ as the instruction and answer tokens in each turn before the current prediction token $\mathbf{n}_a$. For sequences of length $L$, we obtain the target answers generating probability $\mathbf{n}_a$ by:

$$p(\mathbf{n}_a | \mathbf{x}, \mathbf{x}', \mathbf{m}', \mathbf{n}_{instruct,<i}, \mathbf{n}_{a,<i}). \tag{1}$$

In this formulation, $\mathbf{x}, \mathbf{x}'$, and $\mathbf{m}'$ are anchored across all answers. For the sake of clarity, we omit $\mathbf{n}_{system-message}$ and all *<STOP>* tokens, even though they are also taken into consideration in the conditioning. For the model training, we consider a two-stage instruction-tuning procedure. ① *Feature Alignment.* We keep the protein encoder, molecule encoder, and LLM weights frozen, and maximize the likelihood of Equation 3.3 with trainable parameters $\theta = \{\mathbf{W}_p, \mathbf{W}_m\}$. In this way, protein and molecule features $\mathbf{z}_p, \mathbf{z}'_p, \mathbf{z}_m$ can be aligned with the pre-trained LLM word embedding. ② **Parameter Efficient Fine-tuning.** We adopt LoRA (Hu et al., 2021) for the training. The LoRA is an efficient training strategy that maintain high model quality without introducing any delay during inference or necessitating a reduction in the input sequence length. As a result, we keep the visual encoder and the weights of the Large Language Model (LLM) frozen, updating the two projectors $\{\mathbf{W}_p, \mathbf{W}_m\}$ and the LoRA parameters ($\phi$) in the LLM; *i.e.*, the trainable parameters are $\theta = \{\mathbf{W}_p, \mathbf{W}_m, \phi\}$.

**Techinical Details.** Pretraining: We adhered to the official training guidelines of LLaVA, employing the Adam optimizer with an initial learning rate of $5 \times 10^{-5}$, which gradually decreases following a cosine annealing schedule. Our batch size was set at 128, and we trained the two projectors for 5 epochs. LoRA Fine-Tuning: For fine-tuning with LoRA, we set $r = 128$

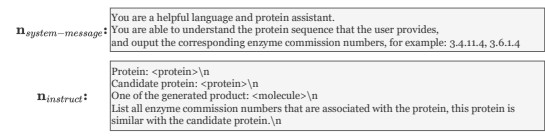

Figure 4: The multimodal instruction used during `LLaPA` LoRA fine-tuning.

and $\alpha = 256$. The learning rates were $5 \times 10^{-5}$ for the two projectors and $2 \times 10^{-5}$ for LoRA modules with the same batch size 128 but 10 epochs. LoRA was applied across all linear modules of LLMs, including [*down_proj, up_proj, q_proj, v_proj, k_proj, o_proj, gate_proj*]. The pretraining and fine-tuning of `LLaPA` were conducted on 8 NVIDIA A6000 GPUs. Retrieval: During training, if protein retrieval fails (i.e., an available protein reference cannot be retrieved), we default to using the query protein sequence as the retrieved protein. Similarly, for missing molecule retrievals, we return zero vectors as the retrieval result. The retrieval database during training is the training set itself to maintain fair comparison with baselines. During inference, we use the original dataset as the retrieval base (including 220K protein sequences filtered from the Swiss-Prot database). For those proteins with multiple EC numbers, we will randomly select one of them for molecule retrieval. Predicting, The format of `LLaPA` is designed for easy reformulation, allowing users to substitute the placeholder "Z" with a period ("." ) to revert to the original, more user-friendly format for reading. Fune-tuning Instruction: We use the instruction for better multimodal optimization during the fine-tuning stage, we leave the system instruction and multimodal instruction in Figure 4.

# 4 EXPERIMENTS

In this section, we first introduce the experimental setup (Section 4.1), then show `LLaPA`'s advance performance (Section 4.2), and finally show indepth analysis about `LLaPA` (Section 4.3).

## 4.1 EXPERIMENTAL SETUP

In this section, we introduce our experimental setup in terms of datasets, evaluation metric, evaluation task, and baselines.

**Datasets.** We selected the Swiss-Prot database (Boutet et al., 2007) as the source of our training data, a subset of the extensive UniProt dataset known for its thorough human review and curated annotations. Employing the data filtering approach described in (Yu et al., 2023), we initially secured approximately 220K protein sequences. Subsequently, we clustered and subsampled these sequences using mmseq2 (Steinegger & Söding, 2017), applying sequence identity cutoffs of 70% to effectively filter out homologous sequences. Our assessment of the `LLaPA` model's competency in predicting EC numbers was performed across four well-regarded benchmarks: (1) `New-392` (or `New`) (Yu et al., 2023), which includes 392 enzyme sequences that span 177 distinct EC numbers. (2) `Price-149` (or `Price`), a collection of protein sequences that were found to be inaccurately or inconsistently labeled in reputable databases like the Kyoto Encyclopedia of Genes and Genomes (KEGG) by automated annotation methods. These sequences were later annotated with labels validated through experiments by Price et al. (2018b). (3) `Multi` (Yu et al., 2023), a dataset comprising enzymes associated with rare EC numbers, each represented no more than five times, with the dataset including over 3,000 samples and covering over 1,000 distinct EC numbers. 4. `Halogenase` (Yu et al., 2023), a dataset that encompasses various halogenases, either marked as uncharacterized and/or hypothetical proteins in UniProt or bearing conflicting annotations in scholarly literature. Through meticulous expert curation and subsequent experimental validations, all halogenase in the dataset were confidently annotated with EC numbers. The sequence identify between training set and testing set `Halogenase`, `Multi`, `New-392`, and `Price-149` are 39.20%, 58.96%, 48.41%, and 42.66%, respectively. Therefore, the performance improvement in `Halogenase` and `Price-149` can indicate the generalization enhancement.

**Evaluation Metric.** Initially, we utilize the F-1 score to compare the performance of `LLaPA` against other baseline models. Subsequently, to delve deeper into the predictive behavior of `LLaPA`, we examine its performance using two different types of accuracy measures.:

$$
\begin{aligned}
\texttt{Acc-1} &= \frac{1}{N} \sum_{i=1}^{N} \frac{number\ of\ true\ positive}{number\ of\ true\ labels} \\
\texttt{Acc-2} &= \frac{1}{N} \sum_{i=1}^{N} \frac{number\ of\ true\ positive}{number\ of\ predicted\ labels}
\end{aligned},
\tag{2}
$$

where `Acc-1` represents the ratio of correct predictions to the total number of ground truth instances, and `Acc-2` denotes the ratio of correct predictions to the total number of predicted EC numbers. The former metric assesses the model's ability to accurately identify the correct EC numbers, while the latter evaluates the model's tendency to predict as many EC numbers as possible.

**Tasks.** We consider two kinds of tasks, one for Full EC number prediction, which needs to predict the four digital numbers, and requires the modal to identify the specific catalytic reaction utilized by the protein, and another is to predict the first three digital numbers of EC numbers that require to understanding the general understanding of the type of reaction the enzyme catalyzes. While it lacks the specificity of the full EC prediction, this broader categorization can be beneficial for tasks like metabolic pathway analysis, where understanding the general role of enzymes can help in mapping out the interconnections and flow of biological processes.

**Baselines.** To highlight the exceptional performance of `LLaPA`, we benchmark it against three state-of-the-art (SOTA) methodologies: (1) For classification, we employ ESM-2 (Lin et al., 2022), a leading general-purpose protein language model. We fine-tune ESM-2 using our training data and then validate its performance across four benchmarks; (2) In terms of retrieval methods, we utilize CLEAN (Yu et al., 2023), which leverages triplet loss to differentiate proteins across enzyme substrate classes; (3) For a translation-based approach, we examine BioTranslator (Xu et al., 2023a), distinguished by its zero-shot learning capability across multiple applications. **Comparison with Structure-Based Protein Predictors:** We used RSCB and AlphaFold2 to construct protein structures from our training data. Since 1% of the proteins in the training set do not have structures, we excluded

Table 1: The comparison of `LLaPA` with the state-of-the-art EC number prediction tools.

| | Halogenase | | | Multi | | | Price | | | New | | |
|---|---|---|---|---|---|---|---|---|---|---|---|---|
| | Acc-1 | Acc-2 | F1 | Acc-1 | Acc-2 | F1 | Acc-1 | Acc-2 | F1 | Acc-1 | Acc-2 | F1 |
| Full EC Numbers | | | | | | | | | | | | |
| ESM2-650M (ft) | 0.0146 | 0.5000 | 0.0155 | 0.3522 | 0.0004 | 0.0412 | 0.4965 | 0.0002 | 0.0403 | 0.5958 | 0.0003 | 0.0276 |
| ESM2-650M (lora) | 0.2162 | 0.0001 | 0.0367 | 0.5975 | 0.0006 | 0.1054 | 0.4406 | 0.0002 | 0.0275 | 0.5375 | 0.0003 | 0.0205 |
| ESM2-650M (linear) | 0.1351 | 0.5556 | 0.1577 | 0.0063 | 1.0000 | 0.0084 | 0.0063 | 1.0000 | 0.2322 | 0.0146 | 0.5000 | 0.0155 |
| BioTranslator | 0.1081 | 0.0571 | 0.0293 | 0.2131 | 0.1625 | **0.1536** | 0.0604 | 0.0448 | 0.0240 | 0.1020 | 0.0802 | 0.0503 |
| CLEAN | 0.1622 | 0.1622 | 0.2140 | 0.0686 | 0.1967 | 0.0951 | 0.0592 | 0.0604 | 0.0958 | 0.0696 | 0.0893 | 0.0475 |
| GearNet | 0.1622 | 0.1622 | 0.2140 | 0.0686 | 0.1967 | 0.0951 | - | - | - | 0.0696 | 0.0893 | 0.2423 |
| GearNet-ESM | 0.1923 | 0.2778 | 0.1667 | 0.0339 | 0.0132 | 0.0161 | - | - | - | 0.2423 | 0.1935 | 0.2406 |
| LLaPA | 0.3514 | 0.3514 | **0.3843** | 0.1414 | 0.1571 | 0.1399 | 0.3423 | 0.3423 | **0.3254** | 0.5016 | 0.5040 | **0.4367** |
| First Three EC Numbers | | | | | | | | | | | | |
| ESM2-650M (ft) | 0.2703 | 0.4545 | 0.2806 | 0.8529 | 0.9667 | 0.8627 | 0.5973 | 0.8558 | 0.6296 | 0.6817 | 0.8193 | 0.7241 |
| ESM2-650M (lora) | 0.2703 | 0.0021 | 0.0562 | 0.1471 | 0.0014 | 0.0140 | 0.6376 | 0.0052 | 0.0898 | 0.5363 | 0.0041 | 0.0414 |
| ESM2-650M (linear) | 0.0811 | 0.5000 | 0.1216 | 0.7353 | 0.9615 | 0.7500 | 0.4161 | 0.9394 | 0.4703 | 0.4336 | 0.8317 | 0.4746 |
| BioTranslator | 0.0811 | 0.0682 | 0.0266 | 0.1311 | 0.1143 | 0.0733 | 0.0470 | 0.0380 | 0.0163 | 0.0459 | 0.0382 | 0.0152 |
| CLEAN | 0.3783 | 0.3514 | 0.3550 | 0.6264 | 0.6443 | 0.6580 | 0.9399 | 0.9344 | 0.9100 | 0.7806 | 0.7303 | 0.7740 |
| GearNet | 0.0769 | 0.1250 | 0.0769 | 0.0192 | 0.0227 | 0.0346 | - | - | - | 0.5529 | 0.6985 | 0.5790 |
| GearNet-ESM | 0.1538 | 0.4444 | 0.1538 | 0.0192 | 0.0213 | 0.0346 | - | - | - | 0.6375 | 0.7276 | 0.6358 |
| LLaPA | 0.9770 | 0.9460 | **0.9563** | 1.0000 | 0.7842 | **0.9335** | 0.9732 | 0.9664 | **0.9701** | 0.9770 | 0.9460 | **0.9563** |

these and used the remaining 99% to train GearNet Zhang et al. (2023c) and ESM-GearNet Zhang et al. (2023b). We applied these models to this structured dataset. However, none of the proteins in the Price dataset have structures available in the RSCB and AlphaFold2 databases, and folding all these proteins using AlphaFold2 is too resource-intensive. Therefore, we evaluated GearNet and ESM-GearNet only on the Halogenase, Multi, and New datasets.

For the ESM-2 model, we utilized the ESM2-650M variant and subjected it to three distinct fine-tuning strategies: ESM2-650M (ft), where all parameters were made trainable; ESM2-650M (lora), where LoRA was applied to the query, key, and value layers, in addition to optimizing an additional classification head; and ESM2-650M (linear), which involved optimization of only the classification head. The classification head's output dimension in ESM-2 corresponds to the total number of Enzyme Commission (EC) numbers identified within both the training and testing datasets. We fine-tuned the publicly available BioTranslator model using the same dataset as `LLaPA`, following the recommended hyperparameters from its documentation. The goal was to accurately align full EC numbers ("EC XX.XX.XX.XX") with their respective protein sequences. For performance evaluation, we utilized a standard multi-label classification approach with a threshold of $0.5$ to calculate metrics. Regarding the CLEAN model, we utilized the official implementation and followed the suggested dataset-specific hyperparameters to derive our final results.

### 4.2 `LLaPA` Achieves Superior Protein Enzyme Understanding

Referring to Table 1, it's evident that `LLaPA` significantly outperforms the baseline models in predicting "Full EC Numbers" across three datasets, registering F-1 score improvements of $\{17.03\%, 9.32\%, 38.64\%\}$ on the Halogenase, Price, and New datasets, respectively. However, it's worth noting that BioTranslator surpasses `LLaPA` in the Multi dataset. Despite this, BioTranslator's Acc-1 is substantially higher than its Acc-2, suggesting a tendency to over-predict EC numbers for each protein—a less-than-ideal approach in practical scenarios. In comparison, `LLaPA` demonstrates competitive performance with BioTranslator, maintaining a closer alignment between Acc-1 and Acc-2, which underscores `LLaPA`'s more dependable predictions.

Furthermore, when focusing on the prediction of the "First Three EC Numbers," `LLaPA` consistently surpasses all baselines across every dataset, with F-1 score improvements of $\{60.13\%, 7.08\%, 6.01\%, 18.23\%\}$. Additionally, the notable discrepancy between Acc-1 and Acc-2 within the Multi dataset highlights `LLaPA`'s limitations in this area, suggesting a need for more comprehensive data to better grasp the nuances of enzymes associated with rare EC numbers.

### 4.3 In-depth Analysis and Ablation Study

***Q1***: **What does the EC Number Reformulation bring to performance? `A1`: Generalizability and Reliability** In our ablation study focused on EC Number Reformulation to address **Q1**, we contrast `LLaPA` with its variants: "`LLaPA` (AAA)" and "`LLaPA` without reformulation". In Table 2, we reveal that modifying the original EC number format by replacing the period ("·") with a letter

Table 2: Ablation study on `LLaPA`. "`LLaPA` (AAA)" involves substituting the periods (".") in the standard EC number format with three "A" letters, while "`LLaPA` w/o reformulation" continues to utilize the traditional EC number format. Additionally, the exclusion of molecule retrieval during the inference process is indicated by "`LLaPA` w/o SMILES", the omission of protein retrieval is denoted by "`LLaPA` w/o protein", and "LLaPA w/ Original Vicuna" replaces the fine-tuned LLM with original vicuna model weight.

| | Halogenase | | | Multi | | | Price | | | New | | |
|---|---|---|---|---|---|---|---|---|---|---|---|---|
| | Acc-1 | Acc-2 | F1 | Acc-1 | Acc-2 | F1 | Acc-1 | Acc-2 | F1 | Acc-1 | Acc-2 | F1 |
| Full EC Numbers | | | | | | | | | | | | |
| LLaPA | 0.3514 | 0.3514 | **0.3843** | 0.1414 | 0.1571 | 0.1399 | 0.3423 | 0.3423 | **0.3254** | 0.5016 | 0.5040 | **0.4367** |
| LLaPA (AAA) | 0.1351 | 0.0676 | 0.1906 | 0.1235 | 0.1030 | 0.1437 | 0.2282 | 0.1141 | 0.2572 | 0.4044 | 0.2086 | 0.3651 |
| LLaPA w/o reformulation | 0.0541 | 0.0270 | 0.0852 | 0.1706 | 0.1393 | 0.1446 | 0.1879 | 0.0940 | 0.2084 | 0.3053 | 0.1607 | 0.2892 |
| LLaPA w/o SMILES | 0.0000 | 0.0000 | 0.0000 | 0.0519 | 0.0574 | 0.0536 | 0.0000 | 0.0000 | 0.0000 | 0.0145 | 0.0153 | 0.0247 |
| LLaPA w/o protein | 0.0270 | 0.0270 | 0.0360 | 0.0717 | 0.0749 | 0.0921 | 0.1342 | 0.1342 | 0.1502 | 0.2425 | 0.2429 | 0.2238 |
| LLaPA w/ Original Vicuna | 0.0000 | 0.0000 | 0.0000 | 0.0525 | 0.0574 | 0.0503 | 0.0000 | 0.0000 | 0.0000 | 0.0109 | 0.0123 | 0.0256 |
| First Three EC Numbers | | | | | | | | | | | | |
| LLaPA | 0.9770 | 0.9460 | **0.9563** | 1.0000 | 0.7842 | **0.9335** | 0.9732 | 0.9664 | 0.9701 | 0.9770 | 0.9460 | 0.9563 |
| LLaPA (AAA) | 1.0000 | 0.8378 | 0.9045 | 1.0000 | 0.4276 | 0.7535 | 1.0000 | 0.6544 | 0.7952 | 1.0000 | 0.5640 | 0.7543 |
| LLaPA w/o reformulation | 1.0000 | 0.6305 | 0.8730 | 1.0000 | 0.4891 | 0.7892 | 1.0000 | 0.6689 | 0.8189 | 1.0000 | 0.6305 | 0.8730 |
| LLaPA w/o SMILES | 0.8649 | 0.8649 | 0.8649 | 0.9836 | 0.5984 | 0.7939 | 0.9195 | 0.9128 | 0.9141 | 0.8724 | 0.8151 | 0.8360 |
| LLaPA w/o protein | 0.4595 | 0.4595 | 0.4595 | 1.0000 | 0.7186 | 0.9005 | 0.9799 | 0.9732 | 0.9739 | 0.9821 | 0.9422 | **0.9599** |
| LLaPA w/ Original Vicuna | 0.1351 | 0.1351 | 0.1351 | 1.0000 | 0.5328 | 0.7627 | 0.9866 | 0.9765 | **0.9866** | 0.8112 | 0.7411 | 0.7720 |

significantly enhances the model's ability to generate plausible predictions for the `Halogenase` dataset, thereby indicating an improvement in generalizability. Moreover, we observed a marked reduction in the discrepancy between `Acc-1` and `Acc-2` following the EC number reformulation. This trend was consistent across both "Full EC Numbers" and "First Three EC Numbers" predictions, underscoring an enhancement in the model's reliability. As illustrated in Figure 1 (**A**), within the embedding space, the character "A" is situated further from the numbers compared to the period ("."), and "Z" is even more distant from the numbers than "A". This spatial arrangement in the embedding space suggests that as the distance from these numbers increases, so too do the generalizability and reliability of the model's predictions.

*Q2*: **Why can the EC Number Reformulation improve performance? `A2`: Better feature quality**
In Figure 5, we display UMAP visualizations derived from the EC number features generated by our model. Each EC number label corresponds to the first digit of the EC number, $i.e.$, the reaction class. Then we also calculate the Silhouette Coefficient (s-score) to assess the clustering quality of various EC number formats; a higher Silhouette Coefficient indicates improved clustering quality. The improved clustering quality indicates the feature quality is better. The results show that replacing "." with the letter "A" can improve the cluster quality and replacing "A" with "Z" can further improve the s-score from $0.187$ to $0.301$. The improvement indicates the EC number reformulation possesses a smoother and more clustered latent space with respect to the ground-truth reaction labels. Meanwhile, the cluster quality improvement aligns with the EC number prediction improvement which implies the improved EC number features quality bolsters the model's performance in predicting EC numbers.

*Q3*: **What are retrieved proteins and molecules responsible for? `A3`: Improve the performance**
The comparison of `LLaPA` against its variations, "`LLaPA` without SMILES" and "`LLaPA` without protein", provides us with several key takeaways: ❶ For "Full EC Numbers", it turns out that information about molecules play a starring role, while information on proteins takes the spotlight for nailing the "First Three EC Numbers" predictions. For instance, our dual-layer retrieval engine boosts the F-1 score from virtually nothing ($0.0\%$) and a modest $3.6\%$ to an impressive $38.43\%$. ❷ The difference between `Acc-1` and `Acc-2` stays pretty much the same, indicating that the reliability is brought by the "EC Number Reformulation". ❸ The enhancements we see with our retrieval engine shine brightest with the `Halogenase` and `Multi` datasets. This suggests that the extra info we pull up helps the language model spot and understand proteins it hasn't met before, showcasing the power of additional data in uncharted territory.

Q4: **What is the ideal protein for the protein retrieval engine? `A4`: The homologous sequence matter.** To tackle this question, we zoomed in on how adjusting our protein retrieval database affects our findings. We started by setting sequence identity cutoffs at $10\%$, $30\%$, $50\%$, and $70\%$, creating four sub-datasets at varying levels of protein sequence similarity. Next, we tested these

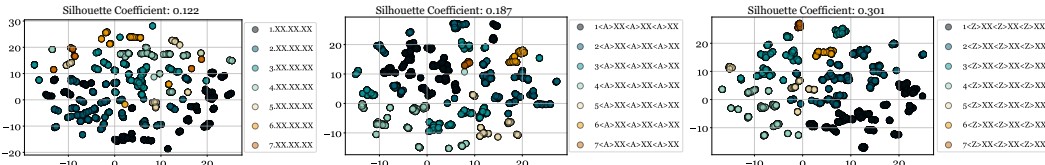

Figure 5: The UMAP visualizations and corresponding silhouette coefficients for the text embeddings of all involved EC numbers in both the training and testing datasets.

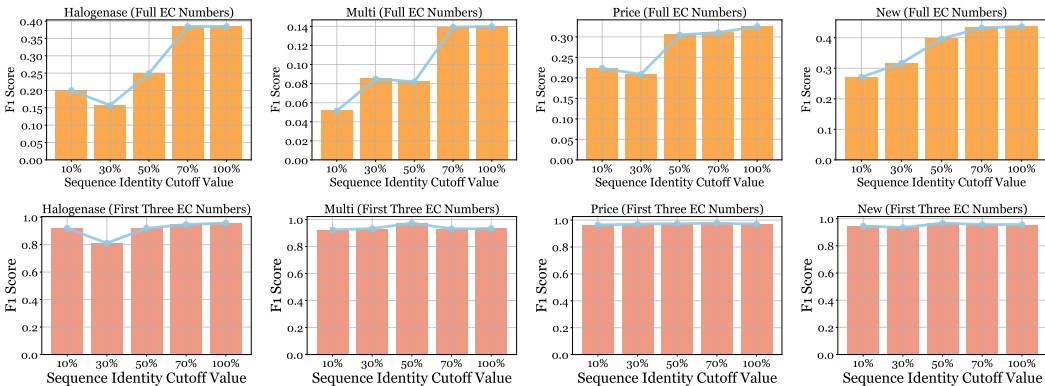

Figure 6: Extra studies about the protein retrieval database. We apply sequence identity cutoff of 10%, 30%, 50%, 70%, crafting five datasets including the original one, to serve as our protein retrieval database throughout the training phase. We've tracked how the F-1 score shifts when we adjust the cutoff values across four datasets, focusing on tasks predicting "Full EC Number" and "First Three EC Number". A higher cutoff value means we're including more homologous protein sequences in our analysis.

sub-datasets to observe any shifts in performance. It's worth mentioning that a 100% cutoff points to our original dataset, detailed in Section 4.1, which includes 220K protein sequences. The 100% cutoff dataset also doubles as our default testing retrieval database. Our strategy involved closely monitoring how the F-1 scores varied with different cutoff values across various datasets, particularly for predicting the "Full EC Number" and the "First Three EC Numbers". What we discovered was quite revealing: incorporating sequences with a higher degree of homology—those closely related protein sequences—proves to be advantageous, especially when tackling the "Full EC Number" prediction tasks. This insight highlights the significance of carefully selecting sequences to enhance the precision of our predictions.

## 5 CONCLUSION

This paper introduces `LLaPA`, a multi-modal framework developed to predict enzyme functions by assigning Enzyme Commission (EC) numbers to protein sequences. Our work represents a pioneering effort to synergize natural language priors (where punctuation such as "." in numbers can resemble large numerical values to LLMs due to their proximity in the word embedding space) and biological priors (emphasizing the evolutionary conservation of functionally critical regions within protein sequences and the catalytic reactions of the corresponding enzymes) in a unified approach using multi-modal large language models.

As a result, `LLaPA` achieves state-of-the-art performance across four public benchmarks, demonstrating its superiority. This underscores the significance of the EC number format and suggests a promising method for integrating biological insights through retrieval mechanisms with LLMs to enhance our understanding of protein enzyme functions. Future directions include a broader exploration of protein function, the integration of our proposed retrieval engine with reasoning capabilities to further augment retrieval effectiveness, and propose large scale of protein dataset with secondary structure then compare with structure-based protein predictors.

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

## A   LIMITATION

Our method primarily utilizes datasets that are publicly available, and our validation process does not include wet lab experiments. This limitation confines the scope of our prediction results. To achieve a more comprehensive understanding and validation of our findings, future work should consider incorporating experimental data from wet lab experiments. Such integration would not only enhance the reliability of our predictions but also bridge the gap between computational predictions and empirical evidence, potentially leading to more accurate and applicable outcomes in the field.

## B   METHOD DETAILS

---
**Algorithm 1** `LLaPA` Training and Inference Pipeline

---
1: **Input:** Query protein $\mathbf{x}$.
2: **if** Training **then**
3:    **Input:** Query protein label $\mathbf{y}$.
4: **end if**
5: **Output:** Predict EC number $\mathbf{y}$.
6: **Require:** Instruction Template $\mathbf{n}_{instruct}$.
7: **Require:** Protein retrieval database **PRD**.
8: **Require:** Molecule retrieval database **MRD**.
9: # Protein retrieval
10: $\mathbf{x}' \leftarrow$ `ProteinRetrieval`$(\mathbf{x}, \mathbf{PRD})$,
11: # Molecule retrieval
12: **if** Training **then**
13:    $\hat{\mathbf{y}} \leftarrow$ `ECNumberExtract`$(\mathbf{x})$ # got EC number of $\mathbf{x}$
14: **else**
15:    $\hat{\mathbf{y}} \leftarrow$ `ECNumberExtract`$(\mathbf{x}')$ # got EC number of $\mathbf{x}'$
16: **end if**
17: $\mathbf{m}' \leftarrow$ `MoleculeRetrieval`$(\hat{\mathbf{y}})$
18: **if** Training **then**
19:    Get prediction $\bar{\mathbf{y}} \leftarrow$ `LLaPA`$(\mathbf{x}, \mathbf{x}', \mathbf{m}', \mathbf{n}_{instruct})$ and calculate the loss with $\mathbf{y}$ to update the model.
20: **else**
21:    Get prediction $\bar{\mathbf{y}} \leftarrow$ `LLaPA`$(\mathbf{x}, \mathbf{x}', \mathbf{m}', \mathbf{n}_{instruct})$
22: **end if**

---

**Training and Inference Pipeline.**   We show the pseudocode of the training and inference pipeline in Algorithm 1. The `ProteinRetrieval` retrieves similar protein sequence in protein retrieval database **PRD** of give protein $\mathbf{x}$, the `ProteinRetrieval` retrieves molecule $\mathbf{m}'$ that related with EC number $\hat{y}$, and `ECNumberExtract` outputs the EC number that corresponding to the input protein $\mathbf{x}/\mathbf{x}'$. LLaPA receive the query protein $\mathbf{x}$, retrieved protein $\mathbf{x}'$ and retrieved molecule $\mathbf{m}'$ to predict EC number for training or inference. The computation cost for training is arround 18 TFLOPs and the inference is arround 2 TFLOPs if the batch size is 1. In practice, we use 8 A6000 for training (batch size is 128) and a single A6000 for inference.

**The Flow of Data and the training details.**   We show the data flow of `LLaPA` in Figure 7(a) (A). The (B) and (C) in Figure 7(a) show the model details during model training. In the first stage, only two modality-specific projectors participate in training, and in the stage two, these LoRA modules added to the LLM are trained simultaneously with these mode-specific projectors.

The training and the inference details of `LLaPA` in Line 17-21 of Algorithm 1 are like this: modality-specific encoder and projector convert each protein $\mathbf{x}/\mathbf{x}'$ and molecule $\mathbf{m}'$ into sequences of protein tokens and molecule tokens, respectively. We then replace the query protein token sequence $\mathbf{x}$ with the special token <protein> in the format "Protein: <protein>\n" in $\mathbf{n}_{instruct}$ (Figure 4). The

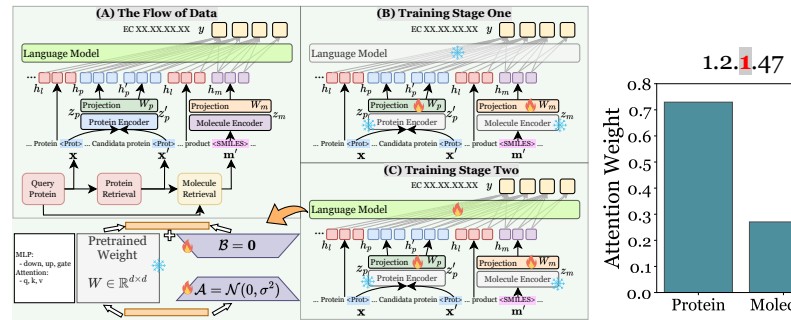

(a) The model details of `LLaPA` include (A) the data flow process, and (B) and (C) the specifics of the two-stage tr--aining procedure.

(b) The attention weight on proteins and molecules when predicting the third digit of an EC number (left) and the last digit of an EC number (right).

Figure 7: The model details of `LLaPA` and the attention weight dynamic from the third to the last digital number of EC numbers.

retrieved protein token sequence, $x'$ is replaced with <protein> in "Candidate protein: <protein>\n" in $\mathbf{n}_{instruct}$. Similarly, the molecule token sequence $\mathbf{m}'$ is replaced with <molecule> in "One of the generated products: <molecule>\n" of $\mathbf{n}_{instruct}$.

Specifically, all text in $\mathbf{n}_{instruct}$ is encoded as $h_l$, and $\mathbf{x}/\mathbf{x}'$ is encoded by protein encoder $E(\cdot)$, and then the output is projected by $\mathbf{W}_p$ to form token sequence $h_p$ and $h'_p$, The special token <protein> in $h_l$ is then replaced by $h_p$ and $h'_p$ Similarly, molecule $\mathbf{m}'$ is encoded by molecule encoder $C(\cdot)$, and then the output is projected by $\mathbf{W}_m$ to form token sequence $h_m$. The special token <molecule> in $h_l$ is replaced by $h_m$. Finally, the LLM uses $h_p, h_m$, and $h_l$ to make predictions.

**Dataset Details.** For the hyperparameters of mmseq2, we used the "mmseqs2 easy-search" command with a sensitivity setting of "-s 5" and a maximum accepted sequences limit of "–max-seqs 10". Default hyperparameters were used for other settings. Our retrieval engine contains $227,363$ proteins and $14,162$ molecules.

# C ADDITIONAL EXPERIMENTS

**Attention Weight Change.** In Figure 7(b), we visualize the attention weights on proteins and molecules when predicting the third and last digits of the EC number. The results show that the attention weight increases when the model predicts the last digit of the EC number, indicating why the molecule contributes significantly to the full EC number prediction.

