# OpenReview forum: "$\texttt{LLaPA}$: Harnessing Language Models for Protein Enzyme Function"
_ICLR.cc/2025/Conference — ICLR 2025 Conference Withdrawn Submission_

### Official Review · Reviewer_Ht2f · 2024-10-28

**Soundness:** 3
**Presentation:** 2
**Contribution:** 2
**Rating:** 3
**Confidence:** 3

**Summary:**

The paper proposes LLM for EC number prediction, namely LLaPA. Other than feeding enzyme and reaction embeddings into some neural networks (such as MLP or CLIP networks), LLaPA projects enzyme and reaction embeddings into language tokens using two projectors, then uses a fine-tuned LLM to predict/retrieval EC numbers.

**Strengths:**

1. Results are really strong.
2. As a protein engineer, having an interactive LLaPA for EC prediction would be great.

**Weaknesses:**

1. Line125-126, typo 'two-tired' -> 'two-tiered'.
2. No codes? Access to the model? (I'd consider to change my score if authors provide the model access/codes)
3. Line 243-246, 'We emphasize that the retrieve logic in the inference phase is reasonable, as proteins with high sequence identify cutoff values typically exhibit similar enzyme functions. Therefore, their molecules in the corresponding chemical enzyme reactions should possess similar catalytic information'. It is a strong statement, you'd better have citations to support you statement.
4. Paper is not easy to follow, too many third-level titles, feel a bit unorganized.
5. As a researcher working on AI for computational biology, I have to say this paper is not novel. Even though the results are really strong (as they claimed), but I dont find the overall approach exciting. Basically, the authors fine-tuned LLMs with new protein-function prompting and trained two language token projectors. LLMs are powerful, but only fine-tuning them for down-streamed tasks lacks novelty and excitement for AI in computational biology.

**Questions:**

Major: No codes? Access to the model? (I'd consider to change my score if authors provide the model access/codes)

---

> ### Author Response · Authors · 2024-11-30
> **Response to reviewer Ht2f (1/1)**
>
> Thanks for your valuable comments. We address your concerns point by point below.
>
> ---
>
> **[W1: Typos]**
> Thanks for pointing it out, we have modified this typo in our revised version.
>
> ---
>
> **[W2 & Q1: Access to code and model]**
> We have already provided our code in the attached **supplementary materials** and will release the complete training set and models upon acceptance to ensure the reproducibility of LLaPA.
>
> ---
>
> **[W3: Citations to support the statement of retrieve logic in the inference phase]**
>
> Thanks for the suggestion. We have added the following citations to support this statement in our revision:
> - [1] Enzyme Function Initiative-Enzyme Similarity Tool (EFI-EST): A web tool for generating protein sequence similarity networks (Biochimica et Biophysica Acta, 2015). This work highlights that high sequence identity often correlates with functional similarity, providing a basis for linking sequence identity with shared enzyme functions.
>
> - [2] Enzyme function prediction using contrastive learning (Science, 2023). This study emphasizes the importance of reliable sequence-function relationships and demonstrates that proteins with similar sequences frequently share functional and catalytic properties. It further validates the utility of retrieval-based approaches for functional annotation tasks.
>
> These references strengthen the claim that proteins with high sequence identity typically exhibit similar enzyme functions, and thus, their associated molecules should share catalytic information.
>
> ---
>
> **[W4: Third-level titles and unorganized content]**
>
> Thank you for the suggestion. We have further addressed the reviewer’s concern by adding an overall illustration in the **Methods** and **Experiments** sections to enhance clarity and make it easier to follow in our revision.
>
> ---
>
> **[W5: Novelty Concern]**
> We respectfully disagree with the reviewer’s assessment. Our LLaPA introduces novel contributions from two key research perspectives:
> - 1. **LLM Perspective**: We address the challenge of predicting EC numbers directly by reformulating the approach to EC number representation. This reformulation significantly enhances the predictive accuracy of LLMs.
>
> - 2. **Biological Perspective**: We develop a two-tiered retrieval engine inspired by biological knowledge. This engine integrates MMseq2 to retrieve relevant proteins, thereby enhancing protein-based predictions, and Rhea to retrieve relevant molecules, improving overall prediction accuracy.
>
> Our approach goes beyond simply fine-tuning the LLM and projectors for specific modalities. The novelty of our work lies in the integration of a protein and molecule retrieval strategy alongside an EC number encoding scheme, both of which contribute to the enhanced performance of EC number prediction.
>
> Additionally, our contributions have been acknowledged by the reviewers. For example, reviewer **kD4L** highlighted our EC number reformulation as “a novel encoding scheme that replaces the radix point with a Latin letter,” while reviewer **fND9** recognized that we “propose a novel solution.” Reviewer **91ka** further identified our two-tiered retrieval engine as “a strong idea.” These acknowledgments validate the novelty and impact of our contributions.
>
> ---
>
> We appreciate the reviewer **Ht2f** time and effort in reviewing our paper. If you have any remaining concerns, please do not hesitate to reach out.

---

> > ### Author Response · Authors · 2024-12-01
> > **Gentle Reminder**
> >
> > Dear reviewer **Ht2f**,
> >
> > We are grateful for your time and review on our work. As the discussion period nears its end, we wish to confirm whether our responses have sufficiently clarified and addressed your concerns, which are listed below.
> >
> > ---
> >
> > - **[W1: Typos]**
> > - **[W2 & Q1: Access to code and model]**
> > - **[W3: Citations to support the statement of retrieve logic in the inference phase]**
> > - **[W4: Third-level titles and unorganized content]**
> > - **[W5: Novelty Concern]**
> >
> > ---
> >
> > We are more than happier to provide additional clarifications before the deadline ends. Please do not hesitate to discuss further concerns.
> >
> > Best,
> > Authors

---

> ### Author Response · Authors · 2024-12-03
> **Follow up Reminder**
>
> Thank you for taking the time to review our work and for your valuable feedback. If everything is clear and you have no further questions or concerns, we kindly ask you to consider adjusting your score. We sincerely appreciate your support and understanding.

---

### Official Review · Reviewer_91ka · 2024-10-29

**Soundness:** 2
**Presentation:** 2
**Contribution:** 2
**Rating:** 5
**Confidence:** 3

**Summary:**

This paper introduces LLAPA, a retrieval-augmented multimodal language model designed to facilitate enzyme EC number prediction. LLAPA retrieves similar protein sequences and related molecular sequences as additional features, augmenting the original protein sequence, and applies a multimodal training approach akin to those used in vision-language models (VLMs). The comprehensive results and analyses presented are good.

**Strengths:**

1. The observation that large language models (LLMs) struggle with predicting numbers and decimal points is intriguing, and the authors propose an innovative solution by examining the embedding space of each character. I find this approach very inspiring.

2. Retrieving similar proteins and molecules to support the classification task is generally a strong idea.

3. The experimental results are solid, and the analysis of each part in the model is comprehensive.

**Weaknesses:**

1. Using related proteins and molecules as additional information for classification tasks is generally a good idea. However, I am concerned about potential data leakage between the retrieval database and test set, which could significantly contribute to the improved performance.

2. LLAPA appears to adopt popular multimodal understanding frameworks (e.g., LLaVa) for EC number prediction tasks. However, I am somewhat unclear on the motivation for using pretrained large language models in this context. It seems feasible to use the retrieved and encoded embeddings as additional features to train a classifier directly, rather than framing the classification task as a dialogue task. Since there doesn’t appear to be any multi-turn or other natural language elements, LLAPA doesn’t seem to function as a protein assistant (e.g., providing enzyme function explanations or reasoning details).

3. Some parts of the presentation, particularly the model details, are unclear.

**Questions:**

1. Please provide more details on LLAPA. For example, how are the MMseqs2 hyperparameters set? How many sequences and molecules do you retrieve, and what is the computational cost for training and inference? These details would enhance our understanding.

2. I would like clarification on the training and inference procedures. In lines 324–328, is the curated dataset used as a training set or solely for MMseqs2 retrieval? How do you prevent data leakage between the MMseqs2-retrieved sequences and the test set? Additional explanation on training and inference would improve clarity.

3. Data leakage could also arise in the molecular retrieval process, as LLAPA uses prior protein knowledge bases and UniProtKB annotations; test set sequences may already appear in these databases. Please add clarifications on this point.

4. The motivation behind using pretrained LLMs and a multimodal training scheme is somewhat unclear. I encourage the addition of baseline comparisons, such as ESM2 + retrieval or the original Vicuna-7b, to illustrate the advantages of pretrained LLMs. It also feels unconventional to frame a classification task as a dialogue. Why not use a linear probe for classification?

5. Some minor typos need correction. For example, in Figure 4, “one of generated…” should be labeled as “<molecule>” rather than “<protein>.”

---

> ### Author Response · Authors · 2024-11-30
> **Response to reviewer 91ka (1/1)**
>
> Thanks for rating our EC number reformulation as “an innovative solution” and acknowledging our retrieval engine as “generally a strong idea”. We provide pointwise responses to your concerns below.
>
> ---
>
> **[W1 & Q2 & Q3: Potential data leakage between the retrieval database and test set]**
>
> Thank you for the reminder. However, we would like to clarify that there is no data leakage problem in our approach. As discussed in **Section 4.1**, the retrieval database and the test dataset correspond to the public training set and testing set in CLEAN [1], which ensures that no proteins appear simultaneously in both the training and testing sets. Consequently, using the training set as the retrieval database does not introduce any data leakage issues, either during training (since the training set in LLaPA is a subset of the training set in CLEAN) or during the retrieval process.
>
> [1] Enzyme function prediction using contrastive learning.
>
> ---
>
> **[W2 & Q4-1: Usage of pre-trained LLM and classification task as a dialogue task]**
>
> As we illustrate in **Section 1**, our goal is to leverage the generalizability of LLMs to enhance performance. Therefore, we begin with a pretrained LLM. Framing EC number prediction as a dialogue task serves two key purposes:
> The pretrained LLM is inherently trained on a self-regression paradigm, making it naturally suited for dialogue tasks.
> The dialogue task avoids restricting the generation context.
> In contrast, using a linear probe for classification requires predefined EC numbers for prediction. This approach limits the model to predicting only EC numbers present in the training set. For instance, while our training set contains 5093 EC numbers, the task requires predicting 5242 EC numbers in total (spanning the training set and four testing sets).
> By adopting the dialogue paradigm, we overcome this limitation, enabling predictions for EC numbers that are absent from the training set. For example, LLaPA successfully predicts the EC number "3.5.1.30," which does not appear in the training set, whereas models relying on linear probes are unable to predict this label. This demonstrates the potential of the dialogue paradigm to generalize beyond the constraints of traditional classification methods.
>
> ---
>
> **[W3 & Q1: More details on LLaPA]**
> - 1. Hyper-parameters of MMseqs2: We use mmseqs easy-search with a sensitivity of -s 5, a maximum accepted sequence count of --max-seqs 10, and the default hyper-parameters for all other settings.
> - 2. Number of proteins for retrieval: 227,363.
> - 3. Number of molecules for retrieval: 14,162.
> - 4. Computation cost: The training process requires approximately 18 TFLOPs, and inference requires around 2 TFLOPs with a batch size of 1. In practice, we use eight A6000 GPUs for training (batch size 128) and a single A6000 GPU for inference.
> Thank you for pointing this out. We have updated this information in our revision (**Appendix B**).
>
> ---
>
> **[Q4-2 Additional baselines]**
> The ESM-2 model does not provide a clear solution for integrating a retrieval engine. Therefore, we use the original Vicuna-7B as an additional baseline LLM. As shown in **Table 2**, "LLaPA with Original Vicuna" performs poorly in the Full EC number setting. In the First Three EC Number settings, it only performs well on the Price dataset. This highlights the necessity of incorporating additional LoRA modules into the LLM and training these modules on our protein datasets.
>
> ---
>
> **[Q5: Typos]**
> Thank you for pointing this out. We have corrected these typos in our revision.
>
> ---
>
> We appreciate the reviewer **91ka** time and effort in reviewing our paper. If you have any remaining concerns, please do not hesitate to reach out.

---

> ### Author Response · Authors · 2024-12-01
> **Gentle Reminder**
>
> Dear reviewer **91ka**,
>
> We are grateful for your time and review on our work. As the discussion period nears its end, we wish to confirm whether our responses have sufficiently clarified and addressed your concerns, which are listed below.
>
> ---
>
> - **[W1 & Q2 & Q3: Potential data leakage between the retrieval database and test set]**
> - **[W2 & Q4-1: Usage of pre-trained LLM and classification task as a dialogue task]**
> - **[W3 & Q1: More details on LLaPA]**
> - **[Q4-2 Additional baselines]**
> - **[Q5: Typos]**
>
> ---
>
> We are more than happier to provide additional clarifications before the deadline ends. Please do not hesitate to discuss further concerns.
>
> Best,
> Authors

---

> ### Author Response · Authors · 2024-12-03
> **Follow up Reminder**
>
> Thank you for reviewing our work and providing valuable feedback. If you have no further questions or concerns, we kindly ask you to consider adjusting your score. We appreciate your support and understanding.

---

### Official Review · Reviewer_fND9 · 2024-11-01

**Soundness:** 2
**Presentation:** 3
**Contribution:** 3
**Rating:** 5
**Confidence:** 4

**Summary:**

The paper presents LLaPA, a LLM framework designed to enhance the prediction of protein function. LLaPA features a dual-level protein-centric retrieval system that retrieves similar protein sequences and relevant molecules, thereby improving EC number prediction. The framework's performance is evaluated on three public benchmarks, demonstrating improvements over existing methods.

**Strengths:**

1. The paper identifies a valid limitation in the prediction of EC numbers by LLMs due to their specific format and proposes a novel solution.
2. The authors effectively demonstrate that the reformulation of EC numbers leads to improved feature quality in Section 4.3, enhancing the model's generalizability and reliability.
3. The framework shows significant performance improvements over existing methods on public benchmarks.

**Weaknesses:**

1. The results indicate a substantial increase in performance due to the dual-layer retrieval engine, but it is unclear how how does performance change when each layer is used separately or in combination? The paper notes that improvements are most significant in the Halogenase and Multi datasets, suggesting that additional data is particularly beneficial for less familiar proteins. How does the retrieval mechanism adapt to different types of proteins, especially those with rare EC numbers or those that are evolutionarily distant from the proteins in the training set?
2. The paper states that information about molecules is crucial for predicting "Full EC Numbers," while protein information is key for "First Three EC Numbers" predictions. A deeper analysis is needed to understand the mechanistic reasons behind this observation. How do molecules and proteins contribute differently to the prediction of full versus partial EC numbers? The comparison with "LLaPA without SMILES" and "LLaPA without protein" variations is insightful. However, the paper should provide a more detailed analysis of the role of SMILES in the context of the model. How does SMILES information integrate with the protein data to enhance predictions?
3. The paper does not provide sufficient evidence that the predicted EC numbers are indeed more accurate due to the proposed method rather than other factors.

**Questions:**

1. The author mentioned that replacing the “.” character improved prediction results, how do you establish a correlation between the proximity of the “.” character to numbers in the embedding space and the difficulty in predicting EC numbers? Is the improvement observed with the replacement of “.” with “A” and “Z” applicable to all types of protein sequences, or are there specific conditions under which it works better? Does this character replacement provide any insights into how the model understands or processes the EC number prediction task?
2. The improvements in F-1 scores are indeed impressive; however, it is crucial to understand whether these improvements are statistically significant. The paper should provide p-values or confidence intervals to substantiate the claim that the observed improvements are not due to chance but are a result of the model's inherent superiority.
3. Could the significant discrepancy between Acc-1 and Acc-2 within the Multi dataset potentially reflect biases or a lack of representativeness in the dataset itself, rather than just limitations of the LLaPA model? Besides Acc-1 and Acc-2, are there other metrics or methods that could be used to assess the model's generalization capabilities when dealing with enzymes associated with rare EC numbers? Besides collecting more data, has the author explored other methods to reduce the discrepancy between Acc-1 and Acc-2?

---

> ### Author Response · Authors · 2024-11-30
> **Response to reviewer fND9 (1/2)**
>
> We sincerely appreciate Reviewer **fND9** for recognizing our EC number reformulation as a "novel solution." Below, we address your concerns point by point.
>
> ---
>
>
> **[W1-1: Performance change when each layer is used separately or in combination]**
> We conducted an ablation study to evaluate the contribution of each component of LLaPA to the overall performance, as shown in **Table 2**. The results highlight the impact of EC number reformulation, protein retrieval, and molecule retrieval on the final performance. This study demonstrates that each module in LLaPA plays a significant role in achieving optimal performance.
>
> ---
>
> **[W1-2: Insights into how the model understands or processes the EC number prediction task]**
> As illustrated in **Figure 5**, our character replacement method enhances the quality of EC number features in the embedding space. This improvement in feature quality reduces the learning difficulty associated with EC number prediction.
>
> ---
>
> **[W1-3: How retrieval mechanism adapt to different types of proteins]**
> Thank you for pointing this out. For different types of proteins, our retrieval process remains consistent. However, there are cases where certain proteins cannot retrieve similar counterparts from our database. In such situations, we use the query protein itself as the retrieved protein and substitute the retrieved molecule with token sequences padded with zeros.
>
> ---
>
> **[W2-1: How do molecules and proteins contribute differently to the prediction of full versus partial EC numbers]**
>
> To address the reviewer’s concern, we have included an analysis of attention weight changes between full and partial EC numbers in our revision (**Appendix C**). The results indicate that molecules contribute more significantly to the final prediction of EC numbers, which provides an explanation for the observed performance decrease.
>
> ---
>
> **[W2-2: How does SMILES information integrate with the protein data to enhance predictions]**
> As explained in **Section 3**, under the "Overview" paragraph, the SMILES and protein data are projected into the LLM's embedding space before being input into the model. Within the LLM, the self-attention mechanism integrates SMILES, protein, and text data, thereby enhancing prediction performance.
>
> ---
>
> **[Q1-1: Correlation between the proximity of the “.” character to numbers in the embedding space and difficulty in predicting EC numbers]**
>
> The correlation is supported by previous works [1,2,3], which suggest that digital numbers positioned closely in the embedding space increase the difficulty of predicting large numbers. Our visualization of digital numbers in **Figure 1 (A)** reveals that the “.” character is positioned close to digital numbers in the embedding space. This proximity suggests that predicting EC numbers is analogous to predicting large numbers, given their similar embedding structure.
> We hypothesize that the proximity of the “.” character to numbers in the embedding space contributes to the increased difficulty of predicting EC numbers. Furthermore, our visualization in **Figure 5** highlights the relationship between prediction feature quality and our EC number reformulation. This reformulation improves the clustering quality of EC number features, which, in turn, reduces the difficulty associated with predicting EC numbers.
>
> [1] Do language embeddings capture scales?
> [2] Methods for numeracy-preserving word embeddings
> [3] Floating-Point Embedding: Enhancing the Mathematical Comprehension of Large Language Models
>
> ---
>
> **[Q1-2: Improvement of replacing “.” with “A” and “Z” for other types]**
> Our training and testing sets collectively encompass over 5,000 distinct EC numbers. Based on this, we believe that replacing the “.” character with either “A” or “Z” is broadly applicable to all types of protein sequences. As demonstrated in **Table 2** and **Figure 5**, our experiments consistently show that replacing “.” with “Z” yields better results than replacing it with “A.”

---

> ### Author Response · Authors · 2024-11-30
> **Response to reviewer fND9 (2/2)**
>
> ---
>
> **[Q2 & W3: Statistical improvement of LLaPA]**
> To address the authors' concerns, we conducted three independent repetitions of the experiments and calculated the p-value between LLaPA and the baseline in terms of the F1 score. The results show that LLaPA achieved statistically significant improvements across all four datasets.
>
>
> |                    |    Halogenase   |      Multi      |      Price      |       New       |
> |--------------------|:---------------:|:---------------:|:---------------:|:---------------:|
> |                    | Full EC Numbers | Full EC Numbers | Full EC Numbers | Full EC Numbers |
> |   ESM2-650M (ft)   |      0.0000     |      0.0000     |      0.0000     |      0.0000     |
> |  ESM2-650M (lora)  |      0.0000     |      0.0000     |      0.0000     |      0.0000     |
> | ESM2-650M (linear) |      0.0000     |      0.0000     |      0.0000     |      0.0000     |
> |    BioTranslator   |      0.0000     |      0.0002     |      0.0000     |      0.0000     |
> |        CLEAN       |      0.0000     |      0.0001     |      0.0000     |      0.0000     |
> |                    | Thee EC Numbers | Thee EC Numbers | Thee EC Numbers | Thee EC Numbers |
> |   ESM2-650M (ft)   |      0.0000     |      0.0000     |      0.0000     |      0.0000     |
> |  ESM2-650M (lora)  |      0.0000     |      0.0000     |      0.0000     |      0.0000     |
> | ESM2-650M (linear) |      0.0000     |      0.0000     |      0.0000     |      0.0000     |
> |    BioTranslator   |      0.0000     |      0.0000     |      0.0000     |      0.0000     |
> |        CLEAN       |      0.0000     |      0.0000     |      0.0000     |      0.0000     |
>
> ---
>
>
> **[Q3-1 Performance discrepancy between Acc-1 and Acc-2 within the Multi dataset]**
> This is an insightful question, and we partially agree with the reviewers' assumption that the performance discrepancy between Acc-1 and Acc-2 could be attributed to the intrinsic characteristics of the Multi dataset. Specifically, in the Multi dataset, each protein is associated with at least two EC numbers, which likely contributes to the significant performance discrepancies observed between Acc-1 and Acc-2 across models.
> However, we do not believe this discrepancy arises from biases or a lack of representativeness in the dataset itself. This is evidenced by the relatively small discrepancies observed for LLaPA in “Full EC Number” (0.0157) and CLEAN in “First Three EC Numbers” (0.0179). Since the training set is consistent across all baselines, we attribute the discrepancy to the methods themselves rather than any inherent bias or representational issue within the Multi dataset.
>
> ---
>
>
> **[Q3-2:Other methods to reduce the discrepancy between Acc-1 and Acc-2]**
> As shown in **Table 2**, our EC number reformulation effectively reduces the discrepancy between Acc-1 and Acc-2. Specifically, by replacing “.” with “A” or “Z,” the discrepancies were reduced from {0.0675, 0.0205, 0.1141, 0.1958} to {0, 0.0157, 0, 0.0004} for the Halogenase, Multi, Price, and New datasets, respectively. To our knowledge, no other technique besides our proposed method has demonstrated the potential to reduce the discrepancy between Acc-1 and Acc-2. We would greatly appreciate any suggestions from the reviewer for alternative approaches to further address this issue.
>
> ---
>
> **[Q3-3: Other metrics or methods to access the model's generalization capabilities]**
> As described in **Section 4.1**, under the paragraph "Datasets," the performance improvements observed in Halogenase and Price-149 datasets indicate an enhancement in generalization. These results demonstrate LLaPA's ability to generalize effectively, particularly on datasets associated with enzymes linked to rare EC numbers.
>
> We agree, however, that incorporating additional metrics or methods to assess the model's generalization capabilities would strengthen our evaluation. If the reviewer could suggest candidate methods to evaluate LLaPA’s generalization performance, we would be happy to implement them.
>
> ---
>
> We appreciate the reviewer **fND9** time and effort in reviewing our paper. If you have any remaining concerns, please do not hesitate to reach out.

---

> > ### Author Response · Authors · 2024-12-01
> > **Gentle Reminder**
> >
> > Dear reviewer **fND9**,
> >
> > We are grateful for your time and review on our work. As the discussion period nears its end, we wish to confirm whether our responses have sufficiently clarified and addressed your concerns, which are listed below.
> >
> > ---
> >
> > - **[W1-1: Performance change when each layer is used separately or in combination]**
> > - **[W1-2: Insights into how the model understands or processes the EC number prediction task]**
> > - **[W1-3: How retrieval mechanism adapt to different types of proteins]**
> > - **[W2-1: How do molecules and proteins contribute differently to the prediction of full versus partial EC numbers]**
> > - **[W2-2: How does SMILES information integrate with the protein data to enhance predictions]**
> > - **[Q1-1: Correlation between the proximity of the “.” character to numbers in the embedding space and difficulty in predicting EC numbers]**
> > - **[Q1-2: Improvement of replacing “.” with “A” and “Z” for other types]**
> > - **[Q2 & W3: Statistical improvement of LLaPA]**
> > - **[Q3-1 Performance discrepancy between Acc-1 and Acc-2 within the Multi dataset]**
> > - **[Q3-2:Other methods to reduce the discrepancy between Acc-1 and Acc-2]**
> > - **[Q3-3: Other metrics or methods to access the model's generalization capabilities]**
> >
> >
> > ---
> >
> > We are more than happier to provide additional clarifications before the deadline ends. Please do not hesitate to discuss further concerns.
> >
> > Best,
> > Authors

---

> ### Author Response · Authors · 2024-12-03
> **Follow up Reminder**
>
> Thank you for taking the time to review our work and provide valuable feedback. If you have no further questions or concerns, we would appreciate it if you could consider adjusting your score accordingly.

---

### Official Review · Reviewer_kD4L · 2024-11-04

**Soundness:** 3
**Presentation:** 3
**Contribution:** 3
**Rating:** 6
**Confidence:** 4

**Summary:**

This paper presents a method called LLaPA, a protein-centric large language model designed to identify protein enzyme functions by predicting EC numbers. It employs a multi-modal approach, reformulating the EC number format within a self-regression framework.

**Strengths:**

To enhance prediction accuracy, LLaPA introduces a novel encoding scheme that replaces the radix point with a Latin letter. Additionally, LLaPA incorporates a two-tiered retrieval engine: (1) at the protein level, it retrieves proteins with similar regions to predict the "First Three EC Numbers"; (2) at the chemical level, it identifies corresponding molecules for the "Full" EC Numbers.

**Weaknesses:**

1. More literature is required. The GearNet and ESM-GearNet are state-of-the-art methods for predicting EC number. GearNet is a geometric pretraining method to learn the protein structure encoder based on the contrastive learning framework. ESM-GearNet learn the joint representation learning on protein sequences and structures. These two methods both employed the structure information, different with the baseline methods in this paper. Although the authors argued that only 6.66% of protein sequences in their training dataset possess corresponding 3D structure, the corresponding 3D structures are easy to achieve by Alphafold2. The authors should introduce GearNet and ESM-GearNet as baseline methods.
2. This paper does not provide sufficient details about the Protein Prior Knowledge Module and the Chemical Reaction Prior Knowledge Module, merely mentioning them without further explanation. To make this clear, a figure or pipline is needed to illustrate what prior knowledge is retrieved and how these modules integrate with the rest of the system.
3. The connection between the multi-modal protein and chemical modules is unclear, as it appears that the models are designed for different targets. It is better to provide a specific example or diagram to show how these modules interact, or how the different targets are reconciled in the final prediction.
4. The architecture of the main LLaPA model should be described in more detail to clarify the flow of data through the model and how different parts of the model are trained.

**Questions:**

Can you feed the protein structure information into a language model?

---

> ### Author Response · Authors · 2024-11-30
> **Response to reviewer kD4L (1/2)**
>
> We are very glad and appreciate that you had a positive initial impression, and we provide respectful and detailed responses to your concerns.
>
> ---
>
> **[W1: More structure-based baselines]**
> Thank you for your suggestion. We used the RCSB and AlphaFold2 databases to construct protein structures from our training data. Since 1% of the proteins in the training set do not have structures available, we excluded these and used the remaining 99% to train GearNet and ESM-GearNet. These models were applied to this structured dataset. However, none of the proteins in the Price dataset have structures available in the RCSB or AlphaFold2 databases, and folding all these proteins using AlphaFold2 is computationally prohibitive. Therefore, we evaluated GearNet and ESM-GearNet only on the Halogenase, Multi, and New datasets. The results in **Table 1** demonstrate that our LLaPA model maintains superior performance.
>
> ---
>
> **[W2: More details about the Protein Prior Knowledge Module and the Chemical Reaction Prior Knowledge Module]**
> Thank you for pointing this out. The Protein Prior Knowledge Module and the Chemical Reaction Prior Knowledge Module correspond to the two-tiered retrieval engine: the protein retrieval engine and the molecule retrieval engine. The pipelines for these modules are depicted in Figure 1(B) and Figure 1(C), respectively.
>
> In the Protein Prior Knowledge Module, we retrieve sequences with high sequence identity, ensuring that functionally critical regions correspond to similar regions. This module incorporates biological prior knowledge by leveraging the conservation of functionally critical regions across high-identity protein sequences. Similarly, in the Chemical Reaction Prior Knowledge Module, we retrieve molecules associated with the enzymatic function of the query proteins. This module integrates biological prior knowledge by utilizing chemical reaction information related to the enzyme function of the query proteins.
>
> Next, our modality-specific encoder and projector transform each protein and molecule into sequences of protein tokens and molecule tokens, respectively. The query protein token sequence xx is replaced with the special token <protein> in the format "Protein: <protein>\n" in $n_{\text{instruct}}$. The retrieved protein token sequence ($x^\prime/x_u^\prime$) is replaced with <protein> in "Candidate protein: <protein>\n" in $n_{\text{instruct}}$. Similarly, the molecule token sequence ($m^\prime/m_u^\prime$) is replaced with <molecule> in "One of the generated products: <molecule>\n" in $n_{\text{instruct}}$.
>
> The text in $n_{\text{instruct}}$ is encoded by the LLM and converted into text tokens. All these tokens are then input into the LLM backbone for training and inference. To further address the reviewer's concern, we have included pseudocode for our training and inference pipeline, along with a diagram illustrating the data flow, in our revised manuscript (see **Appendix B**).
>
> ---
>
> **[W3: Connection between the multi-modal protein and chemical modules]**
> The modality-specific encoders for proteins and molecules are designed for general-purpose applications. Our approach follows the LLaVA training pipeline, which consists of two stages. In the first stage, we train the modality-specific projector layers to map the encoder outputs into the LLM embedding space. In the second stage, we train both the projector layers and the additional LoRA modules integrated into the LLM backbone.
> While the models are tailored for different applications, our learnable modality-specific projectors ensure that the outputs from the encoders are consistently mapped into the same LLM text embedding space. This design enables both modalities to be seamlessly reconciled in the final prediction.

---

> ### Author Response · Authors · 2024-11-30
> **Response to reviewer kD4L (2/2)**
>
> ---
>
> **[W4: More details on main architecture of LLaPA]**
> Thank you for pointing this out. We describe the flow of data and the model training process in **Section 3.3**, specifically in the paragraphs titled “Network Architecture” and “Multi-modal Training.”
> To further address the reviewer’s concern, we have included an additional image in **Appendix B** that illustrates the flow of data and provides more detailed information about the model training process.
>
> ---
>
> **[Q1: Feeding Protein Structure information into a language Model]**
> Thanks for suggesting an interesting and promising idea. To briefly recap, existing structure-incorporating works, such as ESM-GearNet [1], leverage protein sequence embeddings as node features, as demonstrated in serial fusion approaches. Similarly, LM-Design [2] integrates a structure adaptor into the transformer architecture, effectively harmonizing it with sequence embeddings. In our LLaPA framework, to incorporate structural modality into the architecture, we propose leveraging a Structure Encoder alongside the Protein Encoder, using its embeddings as hsh_s (as illustrated in **Figure 1 (D)**, Model Architecture). This approach allows for seamless integration with existing protein and molecular embeddings.
> A key consideration in this process is the necessity of mapped structural information, which may not always be straightforward—for instance, in the case of disordered proteins. Addressing this limitation presents a challenging yet promising avenue for future exploration. In the final version of the paper, we plan to extend the current LLaPA framework to generalize to additional modalities, further enhancing its applicability.
>
> [1] Structure-informed Language Models Are Protein Designers
> [2] Enhancing Protein Language Model with Structure-based Encoder and Pre-training
>
> ---
>
> We appreciate the reviewer **kD4L** time and effort in reviewing our paper. If you have any remaining concerns, please do not hesitate to reach out.

---

> ### Author Response · Authors · 2024-12-01
> **Gentle Reminder**
>
> Dear reviewer **kD4L**,
>
> We are grateful for your time and review on our work. As the discussion period nears its end, we wish to confirm whether our responses have sufficiently clarified and addressed your concerns, which are listed below.
>
> ---
>
> - **[W1: More structure-based baselines]**
> - **[W2: More details about the Protein Prior Knowledge Module and the Chemical Reaction Prior Knowledge Module]**
> - **[W3: Connection between the multi-modal protein and chemical modules]**
> - **[W4: More details on main architecture of LLaPA]**
> - **[Q1: Feeding Protein Structure information into a language Model]**
>
> ---
>
> We are more than happier to provide additional clarifications before the deadline ends. Please do not hesitate to discuss further concerns.
>
> Best,
> Authors

---

> ### Author Response · Authors · 2024-12-03
> **Follow up Reminder**
>
> Thank you for taking the time to review our work and provide valuable feedback. If you have no further questions or concerns, we kindly ask you to consider adjusting your score accordingly.

---

### Author Response · Authors · 2024-11-30
**General Response**

Dear Reviewers,

We extend our sincere gratitude for your thorough review and valuable feedback on our paper. We are truly encouraged by your recognition of the positive aspects of our work, including `strong performance improvement` (Reviewers **fND9**, **91ka**, and **Ht2f**), `novel solution` (Reviewers **kD4L**, **fND9**, and **Ht2f**), and  `protein and molecule retrieval is a generally strong idea` (Reviewer **91ka**).

In addition to addressing your thoughtful comments point-by-point on the OpenReview forum, we have made the following updates to the newly uploaded version of the paper (revisions are highlighted in red):

1. **Additional Structure-based baselines** (Reviewer **kD4L**): Additional baselines results have been added to `Table1`.

2. **Details about the Protein Prior Knowledge Module and the Chemical Reaction Prior Knowledge Module** (Reviewer **kD4L**): Pseudocode for our training and inference pipeline, along with a diagram illustrating the data flow (`Appendix B`) have been included.

3. **Analysis of attention weight changes between full and partial EC numbers** (Reviewer **fND9**): Additional experiments were conducted to assess attention weight changes between full and partial EC numbers, with results included in `Appendix C`.

4. **More details about the retrieval engine on LLaPA** (Reviewer **91ka**): More hyperparameters and detailed settings of our retrieval engine have been added in `Appendix B`.

5. **Additional ablation experiments** (Reviewer **91ka**): Additional ablation experiments that replace LLM with the original Vicuna model were conducted in `Table 2`.

6. **Citations to support the statement of retrieve logic in the inference phase** (Reviewer **Ht2f**): We added citations to clearly support this statement in `Section 3.2`

7. **Adding an overall illustration in the Methods and Experiments sections** (Reviewer **Ht2f**): We added an overall illustration in the Methods and Experiments to make our paper easier to follow in `Section 3` and `Section 4`.

We have made diligent efforts to address all the issues raised and are committed to engaging with any additional inquiries you may have.

Best,
Authors

---

### Note · Authors · 2024-12-16

I have read and agree with the venue's withdrawal policy on behalf of myself and my co-authors.